# Mycoparasitism capability and growth inhibition activity of *Clonostachys rosea* isolates against fungal pathogens of grapevine trunk diseases suggest potential for biocontrol

**Adrienn Geiger**[1‡], **Zoltán Karácsony**[1‡], **József Geml**[2], **Kálmán Zoltán Váczy**[1]*

**1** Food and Wine Research Institute, Eszterházy Károly Catholic University, Eger, Hungary, **2** MTA-EKE Lendület Environmental Microbiome Research Group, Eszterházy Károly Catholic University, Eger, Hungary

‡ AG and ZK share co-first authorship on this work
* vaczy.kalman@uni-eszterhazy.hu

**Data Availability Statement:** All relevant data are available in the paper.

## Abstract

The present study aimed to examine the capability of *Clonostachys rosea* isolates as a biological control agent against grapevine trunk diseases pathogens. Five *C. rosea* and 174 pathogenic fungal strains were isolated from grafted grapevines and subjected to *in vitro* confrontation tests. Efficient antagonism was observed against *Eutypa lata* and *Phaeomoniella chlamydospora* while mycoparasitism was observed to the pathogens of *Botryosphaeria dothidea* and *Diaporthe* spp. pathogens in *in vitro* dual culture assays. The conidia production of the *C. rosea* isolates were also measured on PDA plates. One isolate (19B/1) with high antagonistic capabilities and efficient conidia production was selected for *in planta* confrontation tests by mixing its conidia with the soil of Cabernet sauvignon grapevine cuttings artificially infected with *B. dothidea*, *E. lata* and *P. chlamydospora*. The length and/or the incidence of necrotic lesions caused by *E. lata* and *P. chlamydospora* at the inoculation point were significantly decreased after a three months incubation in the greenhouse on cuttings planted in soils inoculated with the conidia of strain 19B/1, while symptom incidence and severity were unaffected in the case of the pathogen *B. dothidea*. Based on the above results, we consider *C. rosea* a promising biological control agent against some grapevine trunk diseases.

## Introduction

Grapevine trunk diseases (GTDs) encompass a group of infections caused by fungal pathogens that colonize the woody tissues of grapevines, causing discoloration and necrosis [1] in the lignified vascular tissues. Besides the symptoms observed in the vascular system, these diseases also affect the green parts of the plant: chlorosis and necrosis occurrence on leaves, young shoots deformation, and necrotic spots appearance on the berries. In case of long term

**Funding:** This work was financed by the NRDI Fund (projectID: TKP2021-NKTA-16). Kálmán Zoltán Váczy was supported by the János Bolyai Research Scholarship of the Hungarian Academy of Sciences, but the funders had no role in study design, data collection and analysis, decision to publish, or preparation of the manuscript.

**Competing interests:** The authors have declared that no competing interests exist.

infections by Esca disease (one of the most widespread and devastating GTD) the sudden death of the infected part or the whole plant may occur, which phenomenon is called apoplexy [2]. The group of GTDs includes five different syndromes: Black foot disease, caused by *Cylindrocarpon* spp., *Campylocarpon* spp. and *Ilyonectria* spp.; Botryosphaeria dieback, caused by *Botryosphaeriaceae* spp.; Eutypa dieback caused by *Eutypa lata*; Petri and Esca disease, caused by *Phaeomoniella chlamydospora*, *Phaeoacremonium minimum* and several basidiomycetous fungi like *Fomitiporia* spp.; Phomopsis dieback caused by *Diaporthe* spp [3]. The occurrence of GTDs has increased worldwide in the past decades due to the limited use of the efficient preventive and curative techniques. In France, ca. 10% of the productive plants were found to be affected in a 4-year disease incidence estimation of about 700 vineyards [3]. The replacement of dead plants costs around 1.5 billion dollars annually worldwide [4], which is partially due to GTD infections. The highly toxic sodium arsenite has been used to decrease disease frequency, but it is not effective in controlling GTD-related pathogens [5]. Since the ban of this chemical along with other fungicides, there has been no effective way to control GTDs in the European Union. The development of alternative disease management methods, using natural compounds and biological control agents (BCAs) could be a beneficial solution for this problem [1].

To control GTDs, various microorganisms have been tested, including *Bacillus subtilis* [6], *Fusarium lateritium* [7] and *Pythium oligandrum* [8], although *Trichoderma* species are the most widely examined BCAs in this context. Several studies tested *Trichoderma* species for the protection of pruning wounds of grapevine against GTD pathogens [9–11] and some studies focused on root or soil application of *Trichoderma* species [12–14]. Despite the promising results, there are no widely used treatments currently available to protect grafted grapevines from GTD pathogens in nurseries or in the field. Gramaje *et al.* [15] have emphasized the importance of ecological studies searching for potential BCAs against GTDs in the microbiome of grapevine. One way to obtain a new BCA is to find effective antagonists of pathogens. These organisms should be able to function in the same environmental niches as the pathogen they are expected to control. Therefore, it is reasonable to search for BCAs for a given disease from the microbiota of the host plant. A BCA isolated from the host may control multiple diseases caused by pathogens with similar physiology, epidemiology and ecology [16].

*Clonostachys rosea* (syn. *Gliocladium roseum*, teleomorph *Bionectria ochroleuca*) is a soil-borne ascomycetous fungus belonging to the Bionectriaceae family in the Hypocreales order. This species attracts a great attention because of its suitability for biotechnological and pest-control applications [17]. The biotechnological potential of *C. rosea* covers the biotansformation of several molecules like zearalenon [18], the biodegradation of plastics [19] and the production of biofuels [20]. Because its commercial value, the mass production of *C. rosea* was also extensively studied both in liquid [21] and solid state [22] fermentations. Similar to several other hypocrealean fungi, *C. rosea* is known for its antagonistic abilities against numerous plant pathogens, including fungi, nematodes and insects [23, 24]. It can be found globally in soil and decaying plant matter, both in tropical and temperate regions [25] and frequently occurs in grapevine [26–28]. There are only a few studies on the potential use of *C. rosea* as a biocontrol agent against GTDs [29–31]. The above-mentioned studies investigated a limited number of GTD-related pathogens and all of them lack *in planta* investigations. The purpose of the present study was to evaluate the biological control capabilities of this fungus against a wide spectrum of GTD-related pathogens, using *in vitro* and *in planta* experiments.

## Results

### Isolation and identification of fungal strains

We carried out a culture-based investigation of the vascular mycobiota of 100 Cabernet sauvignon grapevines. Beside the several GTD-associated fungal species, five isolates of *C. rosea* were also obtained from plants lacking any internal or external symptoms of GTDs. The morphological characteristics of *C. rosea* isolates were in accordance with the previous description by Schroers [25]. Colonies growing in the dark on PDA medium at 25°C were whiteish, and showed pale orange coloration when grown under fluorecent light. Conidiophores were Verticillium- or Penicillium- like and produced globose conidia.

The identification of the isolates to the species level was done by sequencing of internal transcribed spacer region and additional loci where necessary. Isolates used in the further investigations are listed in Table 1.

### In vitro confrontation tests of *C. rosea* isolates with GTD-related fungi

Growth inhibitions of *C. rosea* strains against the GTD-related pathogens were observed in *in vitro* confrontation tests and the measured percental inhibition rates are summarized in Table 2. The experiments were conducted by culturing the GTD pathogens on PDA plates in the presence or absence of the *C. rosea* isoaltes and measuring their growth rate.

High RGI % values were measured in case of *E. lata* (17.8–29 RGI%) and *P. chlamydospora* (28.8–54 RGI%) species with all the tested *C. rosea* strains. Efficient growth inhibition of these pathogens was accomplished even before the fungal colonies could establish physical contact with *C. rosea* strains (Fig 1A and 1B) indicating the antagonistic activity of the BCA on them. Very low, or no growth inhibition was observed in the case of *B. dothidea*, *Cadophora luteo-olivacea*, and *P. minimum* pathogens (-14.5–29 RGI%). Interestingly, the three *Diaporthe* species tested showed varying responses to *in vitro* confrontation with *C. rosea*. For example, *D. ampelina* mostly showed no change in growth rate, *D. foeniculina* exhibited weak growth inhibition (18.8–22.5 RGI%) with all *C. rosea* strains, while the growth rate of the *D. fukushii* isolate showed positive change with all *C. rosea* strains (-14 to -22 RGI%). This latter is an unexpected result and we consider it worthy of further research, though we could not pursue this direction for this particular paper. We observed overgrowth of *C. rosea* mycelia on the colonies of confrontation partners after a long incubation in case of *B. dothidea* (Fig 1C) and *Diaporthe* spp. (Fig 1D).

Overgrowth rates were measured and summarized in Table 3. The highest values were measured in the case of *B. dothidea* (16.2–18.8 mm) and there was also notable overgrowth above the *Diaporthe* spp. colonies (6.5–18.5 mm). There were no significant differences between the *C. rosea* strains in case of both pathogen taxa.

The ability of the tested *C. rosea* strains to colonize some GTD-related pathogens in the dual cultures suggests mycoparasitism of the affected pathogens. This phenomenon was visually confirmed by microscopic examinations in the case of *B. dothidea* and *Diaporthe* spp. isolates (Fig 2) co-cultured with *C. rosea* isolates on cellophane water agar with a subsequent stainig of viable cells with tetrazolium dye. The tested *C. rosea* strains showed intracellular growth both in *Diaporthe* spp. (Fig 2A) and *B. dothidea* (Fig 2B). Coiling of *C. rosea* around the hyphae of these plant pathogens was also observed (Fig 2C).

### Conidia production of *C. rosea* isolates

The conidia production of the five *C. rosea* strains were measured on colonies growing under artificial light, on PDA plates and summarized in Fig 3. The most efficient conidia producer

**Table 1. List of the isolates used in the present study.**

| Strain number | Species | Sequenced loci | Grapevine trunk disease |
|---|---|---|---|
| 59C/1<br>88C/1<br>99C/1 | *Botryosphaeria dothidea* | ITS, EF | Botryosphaeria dieback [3] |
| 19B/1<br>33C/1<br>89C/3<br>91C/2<br>100C/1 | *Clonostachys rosea* | ITS | no data |
| 33B/4<br>67B/1<br>100B/1 | *Cadophora luteo-olivacea* | ITS | associated with various GTDs [32] |
| 63C/2<br>85B/1<br>98B/1 | *Diaporthe ampelina*<br>*Diaporthe foeniculina*<br>*Diaporthe fukushii* | ITS, EF, ACT | Phomopsis disease [33, 34] |
| T5/2<br>T14/2<br>T15/2 | *Eutypa lata* | ITS | Eutypa dieback [3] |
| T17/5<br>T33/1<br>52B/1 | *Phaeoacremonium minimum* | ITS, ACT | Petri disease, Esca [3] |
| 23A/5<br>37A/3<br>48C/4 | *Phaeomoniella chlamydospora* | ITS | |

ITS: internal transcribed spacer; EF: partial transcription elongation factor 1-α gene; ACT: γ-actin gene

was 33C/1 followed by the 19B/1 strain, with no significant difference in the conidia production of these two strains.

### *In planta* confrontation tests of *C. rosea* 19B/1 isolate with GTD pathogens

The effects of *C. rosea* on the development of GTD symptoms were examined on one year old cuttings. Plants were wounded and inoculated in the xylem by the growing mycelia of *B. dothidea*, *E. lata* or *P. chlamydospora* and grown in a greenhouse in an untreated medium or in soil amended with the conidia of *C. rosea* 19B/1 isolate. Necrotic lesions can be observed in the grapevine cuttings inoculated with GTD-related pathogenic fungi (Fig 4A), while mock-

**Table 2. Growth inhibition of *C. rosea* isolates against GTD pathogens.**

| GTD pathogens | *Clonostachys rosea* isolates | | | | |
|---|---|---|---|---|---|
| | **19B/1** | **33C/1** | **89C/3** | **91C/2** | **100C/1** |
| *Botryosphaeria dothidea* | 9.7 ± 9.7 | 8.8 ± 6.4 | 1.9 ± 1.6 | -1.0 ± 5.0 | 12.9 ± 21.3 |
| *Cadophora luteo-olivacea* | 20.3 ± 15.7 | 8.1 ± 7.5 | 7.9 ±7.3 | 7.1 ± 9.4 | 12.6 ± 14.7 |
| *Diaporthe ampelina* | 3.0 ± 1.4 | 2.0 ± 2.8 | -10.0 ± 2.8 | -6.0 ± 2.8 | 16.0 ± 5.7 |
| *Diaporthe foeniculina* | 13.75 ± 8.8 | 22.5 ± 3.5 | 22.5 ± 3.5 | 18.8 ± 5.3 | 18.8 ± 1.8 |
| *Diaporthe fukushii* | -14.0 ± 8.5 | -14.0 ± 2.8 | -22.0 ± 2.8 | -18.0 ± 2.8 | -18.0 ± 2.8 |
| *Eutypa lata* | 24.8 ± 7.0[ab] | 24.8 ± 1.4[ab] | 17.8 ± 2.2[a] | 19.2 ± 3.8[ab] | 29.0 ± 1.7[b] |
| *Phaeoacremonium minimum* | -14.5 ± 8.2[a] | 3.7 ± 12.6[ab] | 11.4 ± 14.0[ab] | -14.5 ± 13.5[a] | 29.8 ± 0.9[b] |
| *Phaeomoniella chlamydospora* | 38.0 ± 5.6[ab] | 28.8 ± 9.4[a] | 47.1 ± 2.8[b] | 43.1 ± 9.4[ab] | 54.0 ± 2.0[b] |

Mean values of the growth inhibition rates (RGI %) of GTD-related pathogens by *C. rosea* isolates in dual cultures ± standard deviations. Significantly (p< 0.05) differing values of different *C. rosea* strains measured -by Tukey's HSD test- in the case of the same pathogen were indicated as significance groups.

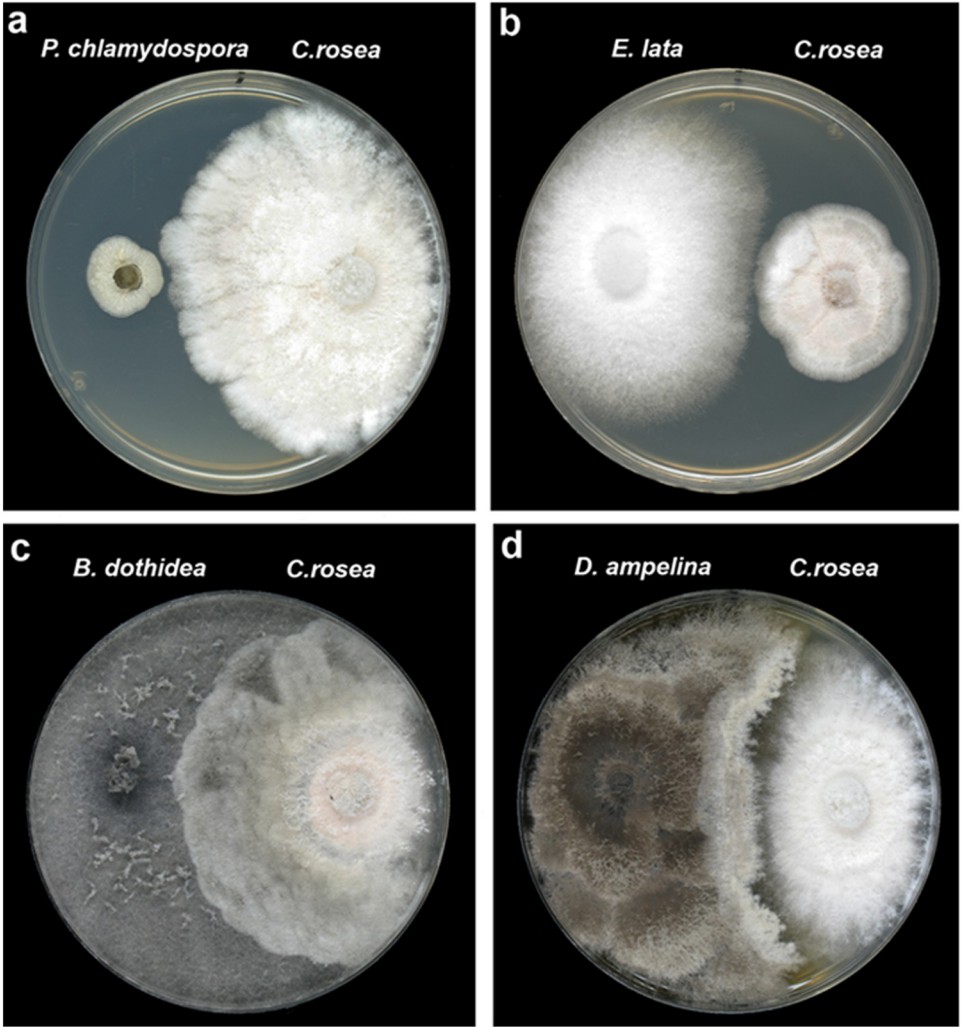

**Fig 1. *In vitro* confrontation tests of *C. rosea* 19B/1 strain with GTD pathogens growing on potato dextrose agar medium.** (**a**) *P. chlamydospora*, 15 days post inoculation (dpi); (**b**) *E. lata*, 8 dpi; (**c**) *B. dothidea*, 15 dpi; (**d**) *D. ampelina*, 15 dpi.

inoculated plants did not show this symptom. Necrotic regions developed deep in the woody tissues in case of *B. dothidea* and *E. lata*, while the tested *P. chlamydospora* isolate necrotized the woody tissues just under the bark of the cuttings (Fig 4A). The length of the necrotic

**Table 3. Mycoparasitism of GTD pathogen colonies by *C. rosea* isolates.**

| | *Clonostachys rosea* isolates | | | | |
|---|---|---|---|---|---|
| **GTD pathogens** | **19B/1** | **33C/1** | **89C/3** | **91C/2** | **100C/1** |
| *Botryosphaeria dothidea* | 17.2 ± 1.9 | 18.8 ± 0.8 | 17.3 ± 2.5 | 16.5 ± 2.6 | 16.2 ± 2.8 |
| *Diaporthe ampelina* | 6.5 ± 0.5 | 11.5 ± 0.7 | 12.0 ± 2.8 | 15.5 ± 2.1 | 13.3 ± 0.4 |
| *Diaporthe foeniculina* | 17.2 ± 1.91 | 18.0 ± 1.4 | 10.5 ± 0.7 | 16.0 ± 1.4 | 18.5 ± 0.7 |
| *Diaporthe fukushii* | 10.5 ± 0.7 | 10.5 ± 0.7 | 5.5 ± 0.8 | 12.5 ± 0.7 | 14.3 ± 1.0 |

Mean overgrowth of the *C. rosea* strains (mm) above the colonies of some GTD-related pathogens ± standard deviations. The overgrowth was measured 19 days post-inoculation. Significances of differences between the *C. rosea* isolates were determined by Tukey's HSD test, resulting in no significant differences between them.

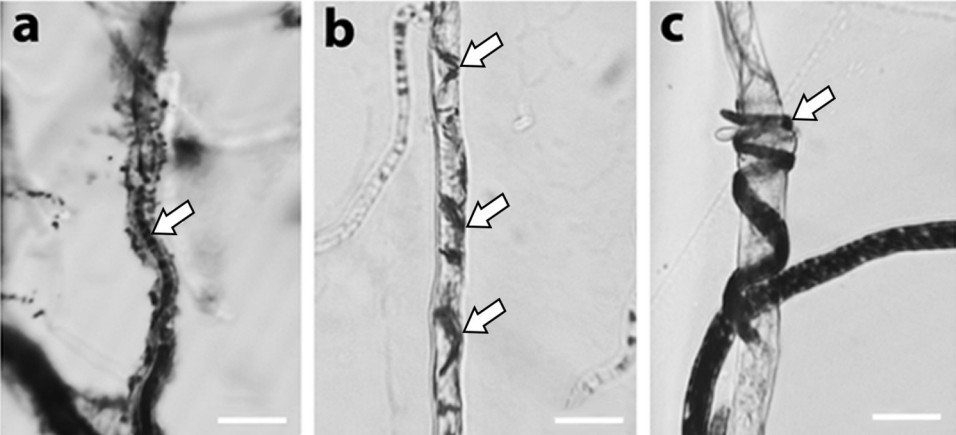

**Fig 2. Microscopic examination of confrontation zones between *C. rosea* 100C/1 strain and GTD pathogens.** Parasitism of *C. rosea* on *D. ampelina* (**a**) and *B. dothidea* (**b,c**) hosts after staining with 5 mM MTT for one hour. Arrows mark *C. rosea* mycelia. Scalebars represent 10 μm.

lesions caused by *E. lata* and *P. chlamydospora* were significantly decreased in case of cuttings planted in *C. rosea*-amended soil, while the disease severity was unaffected by the BCA in case of plants infected with *B. dothidea* (Fig 4B). The re-isolation of *C. rosea* 19B/1 from cuttings not inoculated with a pathogen showed a declining trend from bottom to top. From the base of the cuttings, the strain was re-isolated from three out of five plants, only one out of five

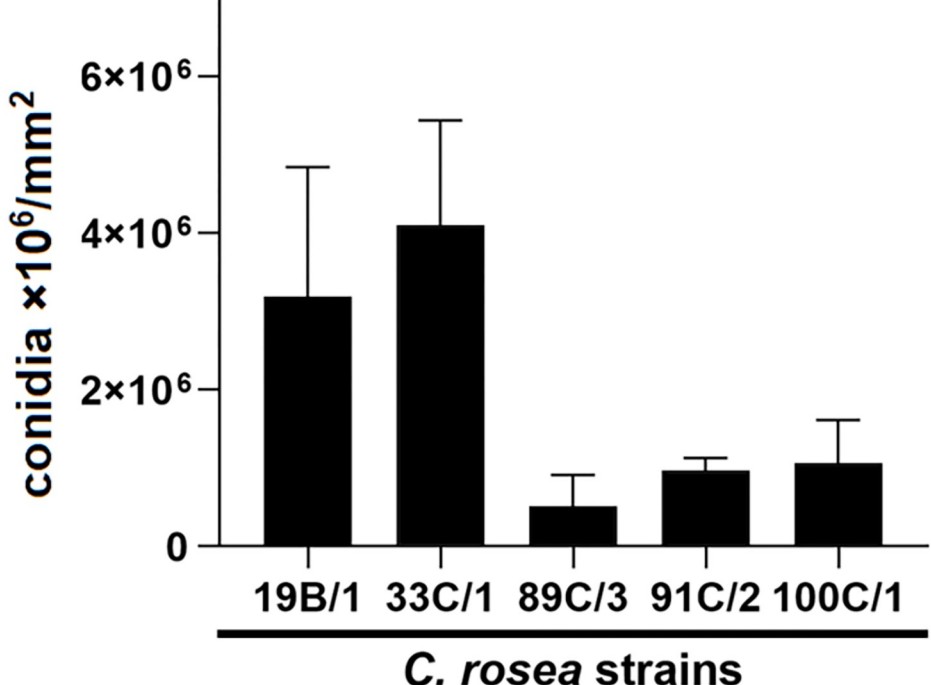

**Fig 3. Comparison of conidia production by *C. rosea* isolates.** Isolates were grown on potato dextrose agar medium for six days under fluorescent light, at room temperature. The numbers of produced conidia were normalized to colony surface.

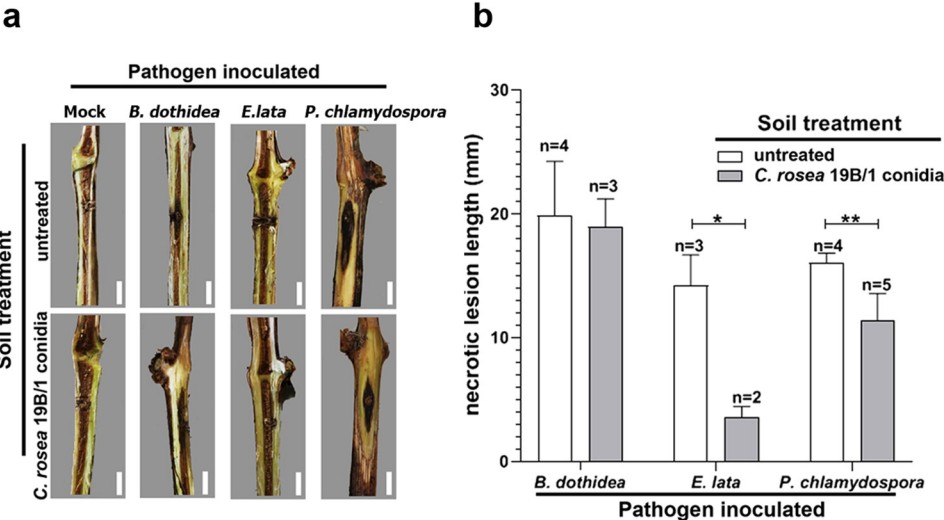

**Fig 4. Effects of *C. rosea* 19/B1 isolate on the development of vascular necrosis caused by GTD pathogens.** (**a**) Representative photographs of wood necrosis developed on Cabernet sauvignon cuttings mock inoculated or infected with *B. dothidea*, *E. lata* and *P. chlamydospora*. Cuttings were grown for 90 days in greenhouse in soil with (upper row) or without (bottom row) $10^4$/g conidia of *C. rosea* 19/B1 strain. Scale bars represent 1 cm. (**b**) Mean lengths and standard deviances of necrotic lesions developed on infected cuttings grown in the untreated or *C.rosea*-amended soils. Numbers above the columns represent the number of symptomatic plants out of five infected cuttings. Asterisks mark significance of differences (* p<0.05, ** p<0.005).

cases at the center of the internode, and it was not found in any of the wound tissue samples. The colony forming unit (CFU) number of the 19B/1 strain increased from $10^4$ to $10^5$ CFU/g in the soil after 90 days of incubation in a greenhouse. It was also notable, that growing mock inoculated cuttings in *C. rosea*-amended medium did not result in any damage on the plants.

## Discussion

Our study demonstrates that *C. rosea* is a potent antagonist of various pathogens causing GTDs, based on the antibiotic and mycoparasitic capabilities of this fungus. Furthermore, the results of the *in planta* confrontation tests suggest, that *C. rosea* can be effectively applied in the soil to prevent the development of GTD-related symptoms on grapevine.

The strains of *C. rosea* tested exhibited particularly strong growth inhibition against *P. chlamydospora* and *E. lata* species. The inhibition was visible even before *C. rosea* established physical contact with the pathogen. This suggests the secretion of antibiotic compounds by the *C. rosea* isolates. Antagonism of *C. rosea* against *P. chlamydospora* was previously shown by Silva-Valderrama *et al*. [31] however antibiosis was not mentioned in that study. Our study is the first report on the antagonistic activity of *C. rosea* against *E. lata*. One possible explanation of this phenomenon could be the previously demonstrated production of antifungal compounds by *C. rosea* [35]. Contrary to the findings of Silva-Valderrama *et al*. [31] on the antibiotic activity of *C. rosea* against the botryosphaeriaceous species *Diplodia seriata* and *N. parvum* our results did not indicate this phenomenon in the case of *B. dothidea*. This may be explained by the physiological polymorphisms of both the pathogenic taxa and *C. rosea*.

Besides the growth inhibition of *P. chlamydospora* and *E. lata*, the observed mycoparasitic behavior of the *C. rosea* isolates on *B. dothidea* and *Diaporthe* spp., as inferred from the intracellular growth and coiling of *C. rosea* hyphae around pathogen hyphae, may suggest different antagonistic strategies of *C. rosea* against different pathogens. The fact, that only the *C. rosea*

cells were stained by the MTT viability dye, suggests the necrotrophic nature of the parasitism. These results are in accordance with previous studies on the mycoparasitism of *C. rosea* on several phytopathogenic fungi, including *Botrytis cinerea* [36], *Fusarium oxysporum* [37], *Sclerotinia sclreotiorum* [38], or even *Trichoderma* spp. [39]. While the mycoparasitic behavior of *C. rosea* is well known and also demonstrated on the GTD-related pathogens *D. seriata* and *N. parvum* [31] our study is the first report of this phenomenon in the case of *Diaporthe* spp.

The observed differences in conidia production among the tested *C. rosea* strains may have implications for their potential use as BCA. The intense sporulation of a fungus used as a BCA is advantageous, because generally conidia are used as inoculum for the treatment of plants, higher sporulation leads to more cost-effective production of a BCA. Efficient spore-producing strains are also likely to be more persistent and intensively distributed in the treated plantations. It is also important, that the compounds responsible for the antibiotic activity of a BCA fungus are usually secondary metabolites. These molecules generally are produced during the stationary phase of fungal growth, alongside with the formation of conidia [40].

Because the causal agents of GTDs are colonizing the vascular tissues of grapevine, the applicability of any BCA against them is strongly dependent on its ability to grow in the tissues of the host plant. The external and internal colonization of host by *C. rosea* was demonstrated previously on cucumber [41] and this fungus was also reported in the tissues of grapevine [26–28]. Our results on the re-isolation of *C. rosea* from the soil and grapevine cuttings showed differences according to the sampling point. The 19B/1 isolate established successfully in the soil and was also able to increase its cell number by about ten-fold. The strain efficiently colonized the woody tissues at the base of the cuttings but was rarely re-isolated from the upper parts of the plants. These results may partly explain the different biocontrol efficacy of 19B/1 strain against the different GTD-related pathogens, observed in case of the *in planta* experiments. The *C. rosea* significantly inhibited the necrosis development in case of *E. lata* and *P. chlamydospora*, which are susceptible to the antibiotic effects of *C. rosea*. This suggests that the antibiotic compounds are secreted by *C. rosea* in the soil, or in the vascular tissues at the base of the cuttings and transported by the xylem sap to the inoculated pathogens. Another possible mode of action which can result in the inhibition of pathogen growth in the absence of direct contact is the triggering of plant defense mechanism by the biocontrol agent. This phenomenon was previously demonstrated in the case of *C. rosea* for example against *B. cinerea* infection tomato [42]. The fact that *C. rosea* was not able to colonize the upper parts of the cuttings explain its ineffectiveness against *B. dothidea* which species is not susceptible to the antibiotic effect of *C. rosea*, but susceptible to mycoparasitism. The latter mode of antagonism requires the establishment of physical contact between the parasite and the prey, which could not be realized in the time period of the experiment due to the relatively slow growth of *C. rosea* in grapevine cuttings. However, field application of *C. rosea* may result in efficient protection against *B. dothidea*, allowing a longer time period for a near-systematic colonization of grapevine woody tissues by *C. rosea*.

Overall, the above results suggest that *C. rosea* has potential as a potent BCA against a wide range of important fungal pathogens of GTDs. The similar results observed in the *in planta* and *in vitro* growth experiments suggest that, in practice, antifungal compounds produced by *C. rosea* may be more important for effective biocontrol purposes than mycoparasitism *per se*, although even the latter could be a promising long-term strategy if near-systemic colonization of grapevine trunks by *C. rosea* can be achieved.

# Materials and methods

## Isolation and identification of fungi from grafted grapevines

One hundred Cabernet sauvignon clone VCR8 grafted on SO4 rootstock were used for the isolation of endophytic fungi. The one year old plants were obtained from an Italian nursery in 2018, shipped bare-rooted, and processed immediately after arrival. The grapevines were symptomless and guarantied to be virus-free. Discs were cut from the graft union, two cm below the graft union, and at the base of the grapevines. The discs were surface sterilized by placing the discs in 70%v/v ethanol, sodium hypochlorite (4%m/v available chlorine), and again in 70%v/v ethanol for two minutes each. The sterilized discs were cut to pieces and placed on potato dextrose agar (PDA) medium amended with 10 μg/ml oxytetracycline to prevent bacterial growth. Plates were incubated at 25°C in the dark. Small portions of emerging mycelia were subcultured to obtain monoclonal isolates. Fungi were identified based on morphological characteristics and by PCR amplification and sequencing of the internal transcribed spacer using ITS1F [43] and ITS4 [44] primers. Where necessary, additional genes were sequenced for unambiguous identification: partial transcription elongation factor 1-α gene with EF-728F and EF-986R primers [45] and/or partial γ-actin gene using ACT-512F and ACT-783R primers [46].

## *In vitro* confrontation tests

To investigate the antagonistic ability of the *C. rosea* strains, confrontation tests were carried out against GTD pathogens. The pathogenic isolates were inoculated individually and in dual cultures with the *C. rosea* strains at five cm distance on PDA medium. For the inoculations 3 mm discs were cut from the edge of fungal colonies growing on PDA medium. The plates were kept in the dark at 25°C. When the pathogen colony approached the *C. rosea* colony to a distance of 5 mm, which happened after 4 to 14 days of incubation, depending on the fastest-growing isolate of the species in question, colony diameter was measured for all cultures of that species and radial growth inhibitions (RGI%) were calculated as described elsewhere [38]. Where observed, the overgrowth of *C. rosea* isolates on the colonies of pathogenic fungi was measured. All experiments were done in triplicates.

## Microscopic investigation of mycoparasitism

In order to visually inspect the mycoparasitism of *C. rosea* on *Botryosphaeria dothidea* and *Diaporthe* spp. (the species where overgrowth of *C. rosea* was observed in confrontation tests), these pathogenic fungi were inoculated in dual cultures with *C. rosea* strains on 2% water agar covered with a piece of sterile cellophane. For the inoculations 3 mm discs were cut from the edge of fungal colonies growing on PDA medium. The plates were incubated at 25°C until physical contact was established between the growing colonies. Confrontation zones were stained with 5 mM MTT (thiazolyl tetrazolium bromide) solution for one hour at 25°C. Stained sections were cut from the cellophane, placed on microscope slides and examined with Alpha BIO-5f (Optika, Italy) microscope, equipped with Artcam-500MI (Artray, United Kingdom) digital camera.

## Quantification of sporulation of *C. rosea*

The conidia production of *C. rosea* isolates were measured by a method described previously [47] with some modification. The tested isolates were mass inoculated on PDA plates with conidial suspensions, in three replicates. The plates were incubated under fluorescent light for eight days at room temperature (21±2°C). After the incubation, six-mm wide agar plugs were

cut from the center of the colonies and suspended in one ml of 0.01%v/v TWEEN 80 solution. The number of the spores was determined with Bürker-chamber. Total conidia numbers were normalized to the colony surface.

### *In planta* confrontation tests

Cabernet sauvignon cuttings were wounded using a drill and inoculated with mycelial plugs (dia. 3 mm) of three GTD-associated pathogen species growing on PDA medium at 25˚C for one week, while sterile agar plugs were placed in the wounds as control. One isolate of *E. lata* (T15/2), *B. dothidea* (99C/1) and *P. chlamydospora* (48C/4) each was used for artificial infections. Two of three pathogenic species (*E. lata*, *P. chlamydospora*) were chosen because they exhibited strong growth inhibition by *C. rosea* and the third (*B. dothidea*) represented a fungus that was highly susceptible to mycoparasitism by *C. rosea*, but showed no growth inhibition. *C. rosea* 19B/1 strain was used for *in planta* confrontation tests. One set of cuttings were planted into soil inoculated with $10^4$ conidia/g of the 19B/1 strain and another set of the cuttings were planted into soil not inoculated with *C. rosea*. Soil was inoculated by mixing with conidial suspension obtained from 10 PDA plates prepared as described in the previous section. Five cuttings were used for each soil × pathogen combination. Plants were incubated in greenhouse with ambient light conditions. Partial control of temperature was achieved by automatically opening top windows activated at elevated temperatures. After 90 days of incubation (from June to August in 2020) the cuttings were uprooted, cut longitudinally after removing the bark, and the length of necrotic lesions was measured. The re-isolation of the *C. rosea* strain 19B/1 was carried out from the inoculated soil and also from three different points from the mock-inoculated cuttings (base, center of internode, wound) grown in the inoculated soil. Serially diluted suspensions of soil were prepared in sterile distilled water and streaked on PDA plates. Re-isolation of fungi from cuttings was done as described above in case of grafted grapevines. Fungi were grown at 25˚C temperature in the dark. The colonies of *C. rosea* were selected according to morphological characteristics and their identity was validated by sequencing the ITS region.

### Statistical analysis

Statistical comparisons were made by GraphPad Prism 5 software demo version (GraphPad Software, San Diego California USA, www.graphpad.com) using Tukey's HSD in case of *in vitro* confrontation tests, one-way ANOVA in case of conidia production measurement and student t-test in case of *in planta* conformation tests. Diagrams were generated with the same software and the layout was edited by Adobe Photoshop CS5 demo version.

## Author Contributions

**Conceptualization:** Zoltán Karácsony, Kálmán Zoltán Váczy.

**Formal analysis:** József Geml.

**Funding acquisition:** Kálmán Zoltán Váczy.

**Investigation:** Adrienn Geiger.

**Methodology:** Zoltán Karácsony.

**Supervision:** József Geml, Kálmán Zoltán Váczy.

**Visualization:** Zoltán Karácsony.

**Writing – original draft:** Adrienn Geiger.

**Writing – review & editing:** Zoltán Karácsony, József Geml.

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
