## [Decision Letter · Decision Letter 0]

7 Jul 2022

PONE-D-22-11451Mycoparasitic and growth inhibition capabilities of Clonostachys rosea isolates against grapevine trunk diseases fungal pathogens suggest potential for biocontrolPLOS ONE

Dear Dr. Váczy,

Thank you for submitting your manuscript to PLOS ONE. After careful consideration, we feel that it has merit but does not fully meet PLOS ONE’s publication criteria as it currently stands. Therefore, we invite you to submit a revised version of the manuscript that addresses the points raised during the review process.

Although this work is important to the readers and researchers of the area but this paper needs significant changes. I recommend the major changes to the paper based on the review reports. Kindly do the required changes and submit a revised paper. I also recommend an English check before submission of the revised version of the paper to the journal.

We look forward to receiving your revised manuscript.

Kind regards,

Vijai Kumar Gupta, PhD in Microbiology

Academic Editor

PLOS ONE

Journal Requirements:

2 Thank you for stating the following financial disclosure:

“This work was financed by the NRDI Fund (projectID: TKP2021-NKTA-16). Kálmán Zoltán Váczy was supported by the János Bolyai Research Scholarship of the Hungarian Academy of Sciences”

Additional Editor Comments:

Although this work is important to the readers and researchers of the area but this paper needs significant changes. I recommend the major changes to the paper based on the review reports. Kindly do the required changes and submit a revised paper. I also recommend an English check before submission of the revised version of the paper to the journal.

Reviewers' comments:

Reviewer's Responses to Questions

**Comments to the Author**

1. Is the manuscript technically sound, and do the data support the conclusions?

Reviewer #1: Yes

Reviewer #2: Yes

2. Has the statistical analysis been performed appropriately and rigorously? 

Reviewer #1: Yes

Reviewer #2: No

3. Have the authors made all data underlying the findings in their manuscript fully available?

Reviewer #1: Yes

Reviewer #2: Yes

4. Is the manuscript presented in an intelligible fashion and written in standard English?

Reviewer #1: Yes

Reviewer #2: No

5. Review Comments to the Author

Reviewer #1: Manuscript entitled “Mycoparasitic and growth inhibition capabilities of Clonostachys rosea isolates against grapevine trunk diseases fungal pathogens suggest potential for biocontrol” mainly talked about C. rosea isolates exhibited promising biocontrol ability against some grapevine trunk diseases. The manuscript is well written, and the results provide useful information for expanding the pathogen range by C. rosea biocontrol. Some points need to be addressed before it can be publication.

1. Line 63-69. Comprehensive review articles should be cited for introducing Clonostachys rosea. e.g. Sun et al. 2020 “Biology and application of Clonostachys rosea”.

2. Line 75 and 232. Identification of fungal strains needs combined morphological characteristics and molecular sequencing.

In line 232, the author declared that the identification of fungal strains would depend on morphological and molecular method. However, in Line 75, the results were mainly focus on ITS amplication. Then what about the morphological characteristics of these fungi? Especially the five C. rosea stains, are their morphological characteristics were same to known C. rosea strains?

3. Table 2. Why the significance analysis was only used for pathogens of Eutypa lata, Phaeoacremonium minimum and Phaeomoniella chlamydospore, but not for other pathogens in this Table?

4. Line 98. Changed “antibiotic” to “antagonistic”.

5. Line 105. Overgrow is a very important phenomenon for mycoparasitism. In this study, the author found the growth rate of Diaporthe strains were nearly not affect by C. rosea. Therefore, how about the overgrow behavior for C. rosea and Diaporthe strains? Just like in Fig.1 c and d, both two strains were overgrowth each other, or only C. rosea overgrowth pathogen strains? From Fig.1, it same like both the two strains are overgrowth each other.

Dose the single inoculation of C. rosea and pathogen strains were set as control in this study?

6. Figure 1, why only (b) E. lata was investigated at 8 dpi, but other pathogens were at 15 dpi?

7. Line 169-172. The sentence needs rewrote as “Our report is the first on the antagonistic activity of C. rosea against E. lata. Contrary to the findings of Silva-Valderrama…”

Confrontation test in plate could reflect the antagonistic activity of C. rosea against pathogens. However, antagonism had several mechanisms, and produce antibiotic is only one of the mechanisms. Therefore, from the confrontation test result, the author could provide conclusion of C. rosea had antagonistic activity against E. lata, but not antibiotic activity. Also the next sentence, C. rosea could produce many compounds with fungicidal activity, is not appropriate.

Reviewer #2: This paper described a study of determining the effectiveness of Clonostachys rosea in controlling fungal pathogens causing grapevine trunk diseases (GTDs). The authors observed significant pathogen growth inhibition in both in vitro and in planta conditions. The reported findings contribute to the collective knowledge of biocontrol agents of C. rosea, especially with regard to in planta conditions. However, the following areas should be mentioned to improve the manuscript:

1. In abstract, experimental data should be provided to support the results and conclusion.

2. The authors identified eight pathogenic species of grapevine trunk diseases basing on the previous researches, therefore, the related references should be supplemented in the Table. Meanwhile, C. rosea strains were also isolated from the plants. We cannot regard them as beneficial microorganisms before a primary test of pathogenicity to grapevine is performed, especially some C. rosea isolates were noticed to significantly promote the growth of some GTD pathogens in vitro test (Table 2).

3. The authors should provide the context when presented the results. It is hard to fully understand the terms used in the results section, however, a short description of the study method and arrangement will help the readers to understand the data presented. Also, the results of Table 2 and Table 3 need to be clearly described in the text.

4. The authors thought the in planta and in vitro experiments suggest “in practice, antifungal compounds produced by C. rosea may be more important for effective biocontrol purposes than mycoparasitism”. It seems arbitrary, for the fungus was mainly found colonizing in soil and the base of the grapevine. In that case, induced resistance to the diseases is more involved, which has been verified in previous studies. It should be mentioned in discussion.

5. Standard three-line tables should be used in the manuscript with the SI units in the tables. The notes should be put below the table, separating with the title.

6. Supplement statistical analysis in the tables.

7. The figures should be improved. Fig. 1, which C. rosea isolate(s) was used? It should be marked. Fig. 2, clarify the arrows and change the bars to a line instead of an area. Fig.3, Rewrite the title to show the research content and put “� 106” in the title of Y-axis. Different colors for the five isolates are not necessary.

8. Necrotic lesions were presented when inoculated with the pathogens (Fig.4), and the authors also mentioned that “while mock-inoculated plants did not show this symptom”. Here, an uninfected cutting should be presented together with the infected cuttings in the figure.

9. In discussion, the authors thought “only the C. rosea cells were stained by the MTT viability dye, suggests the necrotrophic nature of the parasitism”, and “lack of specific structures”. However, different infection structures have been observed in some mycoparasites, including C. rosea.

10. The authors should provide more information about the grapevine samples where the isolates were seperated, for example, the geographical location, cropping years, infected or not, status of soil.

11. In the section of methods, the experimental procedure should be described with more details, therefore, the readers can clearly follow your steps.

12. The term of “sporulation rate” is not correct.

13. The full names of genus should be used when first appear, then only abbreviations are required if not confused.

Furthermore, the authors should consider the following grammatical revisions:

L2-4. I would suggest to change the title to “Mycoparasitism capability and growth inhibition activity of Clonostachys rosea isolates against fungal pathogens of grapevine trunk diseases suggest potential for biocontrol”.

L14. Write the exact number of the pathogens instead of “several”.

L17-18. Change to “to the pathogens of Botryosphaeria dothidea and Diaporthe spp.”.

L19. Change “on in vitro grown colonies” to “on PDA plates”.

L21-22. “by inoculating… into the growth medium” is not consistent with the method.

L31-32. Can be revised to “chlorosis and necrosis occurrence on leaves, young shoots deformation, and necrotic spots appearance on the berries.”

L33-34. There are two mis-spelt words “widespred” and “occour”, and the sentence should be revised to “… the sudden death of the infected part or the whole plant may occur…”.

L47. Change “pest management” to “disease management”.

L81. In Table 1 use “strain number” instead of “strain ID”.

L86. It should be “Growth inhibitions of C. rosea strains …”.

L90. Change “percental radial growth inhibitions” to “growth inhibition rates”.

L115. Change to “Mycoparasitism of …”

L141. Cancel “were”.

L143. Change “developed by” to “caused by”.

L151. Change to “the development of vascular necrosis caused by…”.

L167. Change “This indicates…” to “This suggests…”.

L169-170. Change to “Our study is the first report…” .

L191. Cancel the word “therefore”.

L199. Change to “this fungus was also reported in the tissues of grapevine”.

L207. Cancel the word “species”.

L211. Change to “but susceptible to mycoparasitism”.

L218: Cancel the comma after the word “suggest”.

L251-252. Change “on cellophaned water agar (2 %m/v)” to “on 2% water agar covered with a piece of sterile cellophane”.

L266. Change to “dia. 3 mm”.

6. PLOS authors have the option to publish the peer review history of their article (what does this mean?). If published, this will include your full peer review and any attached files.

Reviewer #1: No

Reviewer #2: No

---

## [Author Response · Author response to Decision Letter 0]

9 Aug 2022

Response to Reviewer#1’s comments

We would like to thank Reviewer#1’s thorough work, important comments and suggestions. We think that the manuscript has significantly been improved with the implementation of the recommended changes done. All changes in the manuscript are highlighted in red. Please find below our responses to Reviewer#1’s comments (in bold).

Reviewer #1: Manuscript entitled "Mycoparasitic and growth inhibition capabilities of Clonostachys rosea isolates against grapevine trunk diseases fungal pathogens suggest potential for biocontrol" mainly talked about C. rosea isolates exhibited promising biocontrol ability against some grapevine trunk diseases. The manuscript is well written, and the results provide useful information for expanding the pathogen range by C. rosea biocontrol. Some points need to be addressed before it can be publication.

1. Line 63-69. Comprehensive review articles should be cited for introducing Clonostachys rosea. e.g. Sun et al. 2020 "Biology and application of Clonostachys rosea".

The characterisation of C. rosea in the introduction was extended in the revised manuscript with a special focus on the biotechnological application of the fungus. Several additional papers were also cited.

2. Line 75 and 232. Identification of fungal strains needs combined morphological characteristics and molecular sequencing. In line 232, the author declared that the identification of fungal strains would depend on morphological and molecular method. However, in Line 75, the results were mainly focus on ITS amplication. Then what about the morphological characteristics of these fungi? Especially the five C. rosea stains, are their morphological characteristics were same to known C. rosea strains?

Yes, the morphological characteristics of the C. rosea isolates were in accordance with its previous description. Related text was added to the manuscript.

3. Table 2. Why the significance analysis was only used for pathogens of Eutypa lata, Phaeoacremonium minimum and Phaeomoniella chlamydospora, but not for other pathogens in this Table?

We used significance analysis to find out if a C. rosea strain is a more potent antagonist of a given pathogen than another C. rosea isolate. All data were analysed in this context, but in case of pathogens other than E. lata, P. minimum and P. chlamydospora there were no significant growth inhibition, or the C. rosea isolates did not differ significantly in the RGI% value.

4. Line 98. Changed "antibiotic" to "antagonistic".

The word is changed according to the suggestion. 

5. Line 105. Overgrow is a very important phenomenon for mycoparasitism. In this study, the author found the growth rate of Diaporthe strains were nearly not affect by C. rosea. Therefore, how about the overgrow behavior for C. rosea and Diaporthe strains? Just like in Fig.1 c and d, both two strains were overgrowth each other, or only C. rosea overgrowth pathogen strains? From Fig.1, it same like both the two strains are overgrowth each other.

Dose the single inoculation of C. rosea and pathogen strains were set as control in this study?

Only the C. rosea strains grow over the Diaporthe spp. and B. dothidea colonies, the opposite can not been observed. Yes, single inoculated controls were also prepared, but have not been presented in the manuscript as figure. We thought that photographs of the dual cultures are appropriate to demonstrate the phenomena we wanted to show, and photographs of single inoculations would not provide additional information.

6. Figure 1, why only (b) E. lata was investigated at 8 dpi, but other pathogens were at 15 dpi?

The two different phenomena we wanted to demonstrate (antibiosis and overgrowth) were expressed at the highest level at different times in case of different pathogens. E.g. P. chlamydospore is very slow growing, therefore it needs more time for its mycelia to grow into the “effective antibiotic range” of C. rosea colonies. In case of B. dothidea and Diaporthe spp. antibiosis have not been detected and for the demonstration of overgrowth, older cultures were more suitable. 

7. Line 169-172. The sentence needs rewrote as "Our report is the first on the antagonistic activity of C. rosea against E. lata. Contrary to the findings of Silva-Valderrama..."

Confrontation test in plate could reflect the antagonistic activity of C. rosea against pathogens. However, antagonism had several mechanisms, and produce antibiotic is only one of the mechanisms. Therefore, from the confrontation test result, the author could provide conclusion of C. rosea had antagonistic activity against E. lata, but not antibiotic activity. Also the next sentence, C. rosea could produce many compounds with fungicidal activity, is not appropriate.

Thank you for the suggestions, the text was rewritten according to your instructions.

Response to Reviewer#2’s comments

We are thankful for the overall positive opinion about our paper and the careful reading of our manuscript. We do believe that the manuscript has significantly been improved during the revision process and fulfills the requirements of publication. All changes in the manuscript are highlighted in red. Reviewer#2’s comments are in bold. 

Reviewer #2: This paper described a study of determining the effectiveness of Clonostachys rosea in controlling fungal pathogens causing grapevine trunk diseases (GTDs). The authors observed significant pathogen growth inhibition in both in vitro and in planta conditions. The reported findings contribute to the collective knowledge of biocontrol agents of C. rosea, especially with regard to in planta conditions. However, the following areas should be mentioned to improve the manuscript:

1. In abstract, experimental data should be provided to support the results and conclusion.

A more detailed description of experimental conditions and results were included in the revised manuscript.

2. The authors identified eight pathogenic species of grapevine trunk diseases basing on the previous researches, therefore, the related references should be supplemented in the Table. Meanwhile, C. rosea strains were also isolated from the plants. We cannot regard them as beneficial microorganisms before a primary test of pathogenicity to grapevine is performed, especially some C. rosea isolates were noticed to significantly promote the growth of some GTD pathogens in vitro test (Table 2).

Suggested references were added to the table. Your concerns on the harmless of C. rosea to grapevines are understandable. However, we can assume the suitability of the application of this fungus on V. vinifera because of the following reasons:

1.) The C. rosea isolates were obtained from healthy plants

2.) There were no symptoms observed in our study on plants growing in media amended with C. rosea.

3.) C. rosea is widely used on several crops without any undesired effects.

Point 1. and 2. were mentioned the revised manuscript.

3. The authors should provide the context when presented the results. It is hard to fully understand the terms used in the results section, however, a short description of the study method and arrangement will help the readers to understand the data presented. Also, the results of Table 2 and Table 3 need to be clearly described in the text.

A short description of the experimental setups were added to the results section in the revised manuscript, and table 2/3 are described with more details in the text.

 4. The authors thought the in planta and in vitro experiments suggest "in practice, antifungal compounds produced by C. rosea may be more important for effective biocontrol purposes than mycoparasitism". It seems arbitrary, for the fungus was mainly found colonizing in soil and the base of the grapevine. In that case, induced resistance to the diseases is more involved, which has been verified in previous studies. It should be mentioned in discussion.

Thank you for the suggestion! The possibility of BCA induced resistance is mentioned in the discussion section with a reference.

5. Standard three-line tables should be used in the manuscript with the SI units in the tables. The notes should be put below the table, separating with the title.

Tables were reformatted according to your suggestions.

6. Supplement statistical analysis in the tables.

Where there were no significant differences, statistical groups were not labelled in the tables. We tried to reflect these cases in the text. 

7. The figures should be improved. Fig. 1, which C. rosea isolate(s) was used? It should be marked. Fig. 2, clarify the arrows and change the bars to a line instead of an area. Fig.3, Rewrite the title to show the research content and put "x106" in the title of Y-axis. Different colors for the five isolates are not necessary.

The suggested changes were done on the figures.

8. Necrotic lesions were presented when inoculated with the pathogens (Fig.4), and the authors also mentioned that "while mock-inoculated plants did not show this symptom". Here, an uninfected cutting should be presented together with the infected cuttings in the figure.

The suggested photographs were added to the figure.

9. In discussion, the authors thought "only the C. rosea cells were stained by the MTT viability dye, suggests the necrotrophic nature of the parasitism", and "lack of specific structures". However, different infection structures have been observed in some mycoparasites, including C. rosea.

Thank you for the suggestion, you are absolutely right. While we in fact did not observe specific infection structures, this does not mean that their lack indicates necrotrophic parasitism. The related sentence was removed from the manuscript. 

10. The authors should provide more information about the grapevine samples where the isolates were separated, for example, the geographical location, cropping years, infected or not, status of soil.

The suggested information was added to the “Materials and methods” section.

11. In the section of methods, the experimental procedure should be described with more details, therefore, the readers can clearly follow your steps.

Additional details were added where it was possible.

12. The term of "sporulation rate" is not correct.

“Sporulation rate” was changed to “conidia production”.

13. The full names of genus should be used when first appear, then only abbreviations are required if not confused.

Corrections were done according to the suggestion.

Furthermore, the authors should consider the following grammatical revisions:

Thank you for the careful reading of our manuscript! All the grammatical corrections suggested below were done and the title was changed.

L2-4. I would suggest to change the title to "Mycoparasitism capability and growth inhibition activity of Clonostachys rosea isolates against fungal pathogens of grapevine trunk diseases suggest potential for biocontrol".

L14. Write the exact number of the pathogens instead of "several".

L17-18. Change to "to the pathogens of Botryosphaeria dothidea and Diaporthe spp.".

L19. Change "on in vitro grown colonies" to "on PDA plates".

L21-22. "by inoculating... into the growth medium" is not consistent with the method.

L31-32. Can be revised to "chlorosis and necrosis occurrence on leaves, young shoots deformation, and necrotic spots appearance on the berries."

L33-34. There are two mis-spelt words "widespred" and "occour", and the sentence should be revised to "... the sudden death of the infected part or the whole plant may occur...".

L47. Change "pest management" to "disease management".

L81. In Table 1 use "strain number" instead of "strain ID".

L86. It should be "Growth inhibitions of C. rosea strains ...".

L90. Change "percental radial growth inhibitions" to "growth inhibition rates".

L115. Change to "Mycoparasitism of ..."

L141. Cancel "were".

L143. Change "developed by" to "caused by".

L151. Change to "the development of vascular necrosis caused by...".

L167. Change "This indicates..." to "This suggests...".

L169-170. Change to "Our study is the first report..." .

L191. Cancel the word "therefore".

L199. Change to "this fungus was also reported in the tissues of grapevine".

L207. Cancel the word "species".

L211. Change to "but susceptible to mycoparasitism".

L218: Cancel the comma after the word "suggest".

L251-252. Change "on cellophaned water agar (2 %m/v)" to "on 2% water agar covered with a piece of sterile cellophane".

L266. Change to "dia. 3 mm".

---

## [Decision Letter · Decision Letter 1]

19 Aug 2022

Mycoparasitism capability and growth inhibition activity of Clonostachys rosea isolates against fungal pathogens of grapevine trunk diseases suggest potential for biocontrol

PONE-D-22-11451R1

Dear Dr. Váczy,

We’re pleased to inform you that your manuscript has been judged scientifically suitable for publication and will be formally accepted for publication once it meets all outstanding technical requirements.

Kind regards,

Estibaliz Sansinenea

Academic Editor

PLOS ONE

Additional Editor Comments (optional):

Reviewers' comments:

Reviewer's Responses to Questions

**Comments to the Author**

1. If the authors have adequately addressed your comments raised in a previous round of review and you feel that this manuscript is now acceptable for publication, you may indicate that here to bypass the “Comments to the Author” section, enter your conflict of interest statement in the “Confidential to Editor” section, and submit your "Accept" recommendation.

Reviewer #1: All comments have been addressed

2. Is the manuscript technically sound, and do the data support the conclusions?

Reviewer #1: Yes

3. Has the statistical analysis been performed appropriately and rigorously? 

Reviewer #1: Yes

4. Have the authors made all data underlying the findings in their manuscript fully available?

Reviewer #1: Yes

5. Is the manuscript presented in an intelligible fashion and written in standard English?

Reviewer #1: Yes

6. Review Comments to the Author

Reviewer #1: (No Response)

7. PLOS authors have the option to publish the peer review history of their article (what does this mean?). If published, this will include your full peer review and any attached files.

Reviewer #1: No

---

## [Editor Report · Acceptance letter]

29 Aug 2022

PONE-D-22-11451R1 

Mycoparasitism capability and growth inhibition activity of *Clonostachys rosea* isolates against fungal pathogens of grapevine trunk diseases suggest potential for biocontrol 

Dear Dr. Váczy:

I'm pleased to inform you that your manuscript has been deemed suitable for publication in PLOS ONE. Congratulations! Your manuscript is now with our production department. 

Kind regards, 

on behalf of

Dr. Estibaliz Sansinenea 

Academic Editor

PLOS ONE